



# Towards Affordable 3D Physics-Based River Flow Rating: Application Over Luangwa River Basin

Hubert T. Samboko[1], Sten Schurer[1], Hubert H.G. Savenije[1], Hodson Makurira[2], Kawawa Banda[3], Hessel Winsemius[1, 4, 5]

[1]Department of Water Resources, Faculty of Civil Engineering and Geosciences, Delft e University of Technology, Stevinweg 1, 2628 CN, Delft, Netherlands
[2]Department of Construction and Civil Engineering, University of Zimbabwe, Box MP 167, Mt. Pleasant, Harare, Zimbabwe
[3]Department of Geology, Integrated Water Resources Management Center, University of Zambia, Great East Road Campus, P.O. Box 32379, Lusaka, Zambia
[4]Deltares, Delft, the Netherlands
[5]Rainbow Sensing, The Hague, the Netherlands

*Correspondence to*: Hubert T. Samboko (hsamboko@gmail.com)

**Abstract.** Unmanned aerial vehicles (UAVs), affordable precise Global Navigation Satellite System hardware, echo sounders, open-source 3D hydrodynamic modelling software, and freely available satellite data have opened up opportunities for a robust, affordable, physics-based approach to monitor river flows. In short, the hardware can be used to produce the geometry. 3D hydrodynamic modelling offers a framework to establish relationships between river flow and state variables such as width and depth, while satellite images with surface water detection methods or altimetry records can be used to operationally monitor flows through the established rating curve. Uncertainties in the data acquisition may propagate into uncertainties in the relationships found between discharge and state variables. Variations in acquired geometry emanate from the different ground control point (GCP) densities and distributions which are used during photogrammetry-based terrain reconstruction. In this study, we develop a rating curve using affordable data collection methods and basic principles of physics. The specific objectives were to: determine how the rating curve based on a 3D hydraulic model compares with conventional methods; investigate the impact of geometry uncertainty on estimated discharge when applied in a hydraulic model; and investigate how uncertainties in continuous observations of depth and width from satellite platforms propagate into uncertainties in river flow estimates using the rating curves obtained. The study shows comparable results between the 3D and traditional river rating discharge estimations. The rating curve derived on the basis of 3D hydraulic modelling was within a 95 % confidence interval of the traditional gauging based rating curve. The physics-based estimation requires determination of the roughness coefficient within the permanent bed and the floodplain using field observation as both the end of dry and wet season. Furthermore, the study demonstrates that variations in the density of GCPs beyond an optimal number (9) has no significant influence on the resultant rating relationships. Finally, the study observes that it depends on the magnitude of the flow which state variable approximation (water level & river width) is most promising to use. Combining stage appropriate proxies (water level when





the floodplain is entirely filled, and width when the floodplain is filling) in data limited environments yields more accurate

discharge estimations. The study was able to successfully apply low cost technologies for accurate river monitoring through

bhydraulic modelling. In future studies, a larger amount of in-situ gauge readings may be considered so as to optimise the

validation process.

**Key words**: Unmanned Aerial Vehicle (UAV), discharge estimation, river Bathymetry, hydraulic modelling


## 1 Introduction

Advancements in technology have led to new opportunities in river monitoring for dam operators, water resource authorities,

environmental agencies and scientists with limited financial capacities (Rafik and Ibrekk, 2001). Hydraulic models play an

important part in river monitoring procedures. However, several different data inputs are required in order to calibrate, validate

and implement hydraulic models. One of the most sensitive of these data inputs is the geometry and bathymetry of a river (Dey

et al., 2019). The geometry is usually described in the form of Digital Elevation Models (DEMs).

DEMs can be generated from a wide range of methods ranging from traditional ground surveying to remote sensing techniques

applied to space- or air-borne imagery. Airborne-based Light Detection and Ranging (LiDAR) systems are capable of

producing highly accurate DEMs (Liu et al., 2008). However, the data has limited spatial coverage and is expensive to acquire

and process. In most cases, traditional ground surveying techniques are laborious, time inefficient, and potentially dangerous

for personnel collecting the data (Samboko et al., 2019).

Space-borne methods provide a non-contact, thus safer, alternative for surveying river terrains. The most common satellite-

based topography data sources are the Shuttle Radar Topography Mission (SRTM) DEM and the Advanced Space-borne

Thermal Emission and Reflection Radiometer (ASTER) DEM. Unfortunately, there is a significant trade off which needs to

be taken into account when applying satellite data for the purposes of river monitoring. Most freely available satellite-based

terrain data sources such as ASTER (15m) and SRTM (30m) do not satisfy the required combination of spatial and temporal

resolution necessary for accurate river monitoring. Consequently, while satellite data is promising for larger rivers, their spatial

and temporal resolution is not appropriate for small to medium rivers (Kim, 2006).

It is within this technological gap that Unmanned Aerial Vehicles (UAVs) platforms equipped with cameras, continue to be

developed and applied due to their relatively low cost, high resolution and efficient application processes. The UAV collects

overlapping images which are geotagged and subsequently merged together using photogrammetry (Skondras et al., 2022).

The photogrammetric process in turn produces a number of outputs which include a digital elevation model (DEM). However,

in order to reconstruct accurate geometries, the photogrammetry process requires Ground Control Points (GCPs) to identify

the precise location of matter in the visible domain (Smith et al., 2015).





The process of applying GCPs is laborious and time consuming, therefore it is important to minimise the number of GCPs collected without significant compromise on accuracy (Martínez-Carricondo et al., 2018; Smith et al., 2015; Woodget et al., 2017). Several studies have been conducted in order to determine the optimal number of GCPs necessary for accurate geometry reconstruction (Awasthi et al., 2019; Coveney and Roberts, 2017; Ferrer-González et al., 2020). Very few studies however, have investigated the impact of uncertainties in geometry on the estimated flow when applied in a 3 D hydraulic model. One

such study conducted by Samboko et al. (2022), investigated the impact of variations in the number of GCPs on the hydraulic conveyance. The study concluded that nine GCPs spread out across 25 hectares to optimally represent the full spectrum of elevation variations is sufficient for accurate conveyance estimation. However, the conveyance is a proxy of actual flow and may not be fully indicative of the actual discharge. Therein lies this research study gap, which seeks to develop a more physics-based rating curve using a combination of low-cost data collection equipment and 3D hydraulic modelling. We assess the

robustness of the method by determining how inaccuracies in the geometry caused by varying GCP numbers, ultimately propagate into stage-discharge relationships. Furthermore, the study investigates how uncertainties in proxies of flow that may be derived from satellite platforms, such as river width (through surface water detection) or water level (from e.g. altimetry missions) propagate into uncertainties in discharge estimation.

The following research questions are investigated to determine whether the mentioned factors have a significant effect on the

accuracy of results.

How does the rating curve produced by a 3 D hydraulic model compare with conventional methods?

How do uncertainties in the surveyed geometry propagate into estimated discharge when applied in a 3D hydraulic model?

How do uncertainties in proxies of flows from satellite data propagate into uncertainties in discharge estimation?


## 2 Material and Methods

In brief, the experiment consist of the following steps: select a suitable study site as far away as possible from impediments which may cause backwater effects and with a relatively straight river profile, use a combination of the UAV, RTK-GNSS,

and ADCP to determine the wet /dry bathymetry and slope, merge the dry and wet bathymetries and subject the merged bathymetry to boundary conditions within a 3D hydraulic modelling environment, determine the roughness coefficient and run the hydraulic model a number of times until a relationship between flow and stage (rating curve) can be determined, compare the rating curve with traditional rating curves then repeat this experiment using varying bathymetries and compare the outputs to determine if there is a significant difference in the results. *Figure 1* presents a schematic of the experiments conducted in

this study.





## 2.1 Data collection methods

A detailed description of how the dry and wet river bathymetry can be collected using low-cost UAV and GNSS device is
introduced in section 2 of a study in Samboko et al. (2022). In short, the method consists of the following steps: an airborne
instrument (e.g. UAV) is used to collect overlapping and geotagged images which are in turn converted into dry bathymetry
through photogrammetry. Ground control points measured using low cost RTK GNSS equipment are used to rectify
inaccuracies in the bathymetry. The wet bathymetry is measured using a combination of an RTK GNSS and an echo sounding
instrument (e.g. fish finder).  The waterline is then measured using the RTK GNSS so as to correct any doming effect which
may be caused by uncertainties in correcting radial lens distortions. Finally, the wet and dry bathymetries are merged through
linear interpolation to form a seamless full bathymetry.

## 2.2 Study Site

The study was conducted in Southern Zambia along the Luangwa River, downstream of the Luangwa Bridge. The Basin has
a catchment area of approximately 160,000 km$^2$. The Luangwa River originates in the Mafinga Hills in the North-Eastern part
of Zambia and is approximately 850 km in length, flowing in South-Western direction (The World Bank, 2010). The river
drains into the Zambezi River, shaping a broad valley along its course, which is well-known for its abundant wildlife and
relatively pristine surroundings (WARMA, 2016). The study area is approximately 25 hectares.

For purposes of comparison, the specific location of the study site is only a few kilometres from the Zambia Water Resources
Management Authority (WARMA) permanent gauging station and a couple of hundred metres from the site where a similar
study based on a 1D Hydrologic Engineering Center - River Analysis System (HEC-RAS) model (Abas et al., 2019). These
sites may be considered similar in their hydraulic conveyance properties, given that they are geographically close to each other
and their geomorphological characteristics are similar. A dataset of discharge and stage measurements, taken by WARMA
between 1948 and 2002 is available for rating curve comparison. We surveyed the flow and water level twice, at the end of
the rainy season and at the end of the dry season so as to capture both low (only permanent channel) and intermediate (also
partly floodplain) flow conditions. *Figure 2a* shows the location of the study site within the Luangwa Basin. *Figure 2b* shows
the location of the study site in relation to the 2 other sites.


### 2.3 Hydraulic Modelling

For hydraulic simulation, we used D-Flow Flexible Mesh (D3DFM) (Deltares, 2020). D3DFM solves the nonlinear shallow water equations in 1D, 2D or 3D or combinations thereof using a flexible mesh domain. Within D3DFM two different layering

methods are provided for 3D models, the sigma ($\sigma$) method and the Z-method. The Z-method is based on the Cartesian Z-coordinate system resulting in straight horizontal coordinate lines. Layers in the $\sigma$-model increase or decrease in thickness as the water depth in the model increases or decreases. The relative thickness distribution of the different layers however remains fixed (Deltares, 2020). *Figure 3* shows how the sigma layers and Z layers differ spatially in thickness.


A hydraulic model consisting of a bed level, a grid structure, mathematical formulations describing the physical processes and corresponding necessary assumptions and approximations requires boundary conditions to simulate the desired hydraulic processes. In case of a river model these boundary conditions do often comprise an inflow and outflow of water implied by a discharge, velocity or water level. In D3DFM models these boundary conditions can be imposed as a time series or as a

harmonic signal.

Besides the boundary conditions, there are initial conditions and physical parameter values to be assigned to the model, for example initial water levels, the water temperature and a uniform friction coefficient. This friction coefficient influences the maximum velocity of the water at the river bed and therefore affects the discharge capacity and water level in the simulation (Saleh et al., 2013). The roughness can be described by different formulations like Chézy, Manning or White-Colebrook which

all contain a certain roughness coefficient that needs to be specified. For the purpose of this study, the Manning coefficient is chosen as it is more applicable to open channels (Zidan, 2015).

### 2.4 Description of Data Requirements for D3DFM

Model setup and evaluation needed the bathymetry, boundary conditions (discharge and water level) and the roughness

coefficient.

*Bathymetry data requirements*

The bathymetry of the terrain is established through merging and volumisation of photogrammetric data with sonar measurements. In brief, the Digital Terrain Model (dry bathymetry) is merged with river transects (wet bathymetry) and

subsequently volumised into a complete seamless bathymetry through linear interpolation. More details on this method can be found in Samboko et al., (2022).

The seamless bathymetry is then cut perpendicular to the flow direction on both sides in preparation for input into D3DFM. *Figure 4* shows an example of a DEM which has been volumised and subsequently cut on both sides.



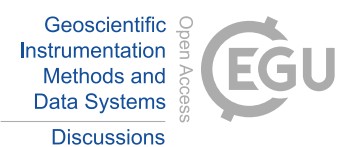


In order to use the point cloud in a model, the area should be extended both downstream and upstream. The extensions is required to ensure that upstream water can numerically spread over the entire width realistically, and downstream to ensure that the imposed downstream boundary does not affect the water levels and velocities in the area of interest. A small selection

of 1200 coordinates over the complete width on each side is taken. This small stretch is reproduced every 36 meters in the direction of flow (or opposite for the extension to the north), this means the longitudinal and latitudinal values are shifted slightly and the height is subtracted or added with the corresponding slope. The point cloud is extended both upstream and downstream with 118 stretches, corresponding to 4248 meters, which is significantly more than the adaptation length (2.1 km). The adaptation length is the distance required to counter the effects of backwater. After volumising the model for the last time,

the final result is a point cloud containing $4.76 \times 10^9$ coordinates representing approximately 9.2 km of the Luangwa River. *Figure 5* shows the elongated bathymetry which is imported into D3DFM representing the bed level.


### 2.5 D3DFM setup, calibration and evaluation

The model was setup with two Manning roughness configurations. One based on the main channel using the dry season observation set (water level, flow and velocimetry) and another where the degrees of freedom are extended to two roughness values (one main channel, one floodplain) using an observation taken during both the wet and dry season observations. This

is to evaluate whether one visit is sufficient, or whether multiple visits are recommended.

In order to determine the optimal roughness coefficient of the main channel in the dry season, we constrained the model through optimisation of a combination of surface velocity and water level. The start value for Manning's friction coefficient was set at 0.018 s/m$^{-1/3}$, the median of the $n$ value (Manning) for sandy straight uniform channels which ranges from 0.012 to

0.026 s/m$^{-1/3}$(Arcement and Schneider, 1989). The upstream boundary condition which was measured in the field was kept constant at 191 m$^3$/s. We imported the coordinates of known surface velocities which were measured using Large Scale Particle Image Velocimetry (LSPIV) and a current meter. Similarly, coordinates of known water levels which were measured using an Acoustic Doppler Current Profiler (ADCP) were imported into the model and compared to the simulated water levels. Note that the use of ADCP could be replaced by the use of a more cost efficient sonar, such as a fishfinder device, to keep the

method entirely affordable. The comparison is based on the Mean Average Deviation (MAD). The score was based on 5 measurements for the current meter and 10 for LSPIV. The simulated water level was similarly assessed with 5 observation





points located in the centre of the wet bathymetry. A combination which yields the lowest values of MAD indicates an optimal roughness coefficient to proceed with.

The second model setup incorporated the wet and dry roughness coefficients. On the main channel, we applied the roughness which had been calibrated in the dry season. On the floodplain, we applied a roughness coefficient of 0.040 s/m$^{-1/3}$ which was derived through a 1 D HEC RAS model in the wet season. A summary of the derivation is describe in *Annex 1*

After the model was constructed and calibrated, the next step was to accurately predict discharges other than 191 m$^3$/s. Establishing a stage-discharge relationship requires rating points (a discharge with corresponding stage) produced by the
model. Hence, the model was run at least 20 times with changing boundary conditions. The upstream boundary condition was given by a discharge ranging from 5 to 3000 m$^3$/s and the downstream boundary condition was determined through repetitive iterations which estimated the water level based on slope. Finally, both models were compared with a traditional rating curve constructed by WARMA. The 95% confidence interval of the WARMA rating curve will be used to generally judge the accuracy of the more physically constructed rating curve. Statistical model evaluation tools, Nash–Sutcliffe efficiency ($E_{ns}$)
and Percentage bias ($P_{bias}$) are also used to determine significant differences among the simulated curves. The selected criteria are recommended for model evaluation because of their robust performance rating of simulating models.(Moriasi et al., 1983). $P_{bias}$ measures the tendency of the simulated data to either under-estimate or over-estimate the observed WARMA readings. Low magnitudes indicate optimal model simulation. $E_{ns}$ indicates how well the plot of observed versus simulated data fits the 1:1 line. NSE and PBIAS are computed as shown in *equation 1* and *equation 2*.


$$E_{ns} = 1 - [\frac{\sum_{i=1}^{x}(O_i - P_i)^2}{\sum_{i=1}^{x}(O_i - O_{mean})^2}] \qquad \text{Eq1}$$

$$P_{bias} = \frac{\sum_{i=1}^{x}(O_i - P_i)}{\sum_{i=1}^{x} O_i} \qquad \text{Eq2}$$

**2.6 Comparison of discharge estimations based on varying geometries**

In order to evaluate the impact of the number of GCPs on the estimated discharge, four elevation models reconstructed based on 5, 9, 13 and 17 GCPs are fed into the D3DFM hydraulic model under similar boundary conditions. The preparation of the bathymetries is similar to that which has been described in section 3.2. We inter-compare the different rating curves





individually to evaluate if there are any notable differences. *Figure 6* shows the varying GCP configurations used in the generation of bathymetries.


Evaluation of the propagation of continuous width and depth observations on uncertainty of discharge estimation
The two main proxies of flow that we assessed, and which potentially can be used for continuous monitoring through satellite observations, are water level and river width. In preparation to measure river width, we placed a cross section perpendicular
to river flow where the cross-sectional must cut across the entire flood plain. *Figure 7* shows the location and orientation of the cross section.

Thereafter, the model is run 20 times with varying upstream boundary conditions between 5 and 3000 m³/s. For each simulated upstream discharge value, we measured and recorded the width in the simulation. After calculating the average river width we
established a discharge versus river width relationship (Q-b). With the assumption that our estimated widths could be +/- 5 meters uncertain, or in even more uncertain cases =/-10 meters, we estimated the river flow and its uncertainty through the established relationships between flow and depth, and flow and width respectively. This allowed us to assess at which point along the full stretch of the floodplain which proxy is more likely to produce accurate discharge estimations. This process was repeated with water depth as the proxy.


## 4 Results and discussion

The impact of photogrammetry-based geometry on the estimated discharge was assessed through three steps: comparing the rating curve of the D3DFM model with traditional methods, comparing rating curves based on geometries constructed using different GCP numbers in D3DFM, and evaluating how the uncertainty in models based on proxies of flow (width and water
level) propagate into discharge inaccuracies.

## 4.1. Comparing the Rating Curve of the D3DFM Model with Traditional Methods

Before the comparison of D3DFM with other models, calibration and validation was performed. The surface flow velocity and the water depth were used to calibrate the model whilst model validation was performed based on a visual assessment of the
RTK tie line and surface velocity. The measured variables are summarised in *Table 1*.





Table 1 The experiments used for models' calibration and validation.

| Phase | Data set | Description | Use |
|---|---|---|---|
| 1 | Surface velocity (LSPIV, Current meter) and Water Depth (ADCP & RTK GNSS) | Determining the Roughness ($n$) coefficient | Calibration |
| 2 | RTK tie line and surface velocity | Testing the models predictive capacity | Validation |

The model setup required calibration of the roughness coefficient based on an optimal combination of the simulated water
surface velocity and water level. The simulated velocities for the different roughness values were compared to the current
meter and LSPIV measurements using the Mean Average Deviation (MAD) and percentage bias. *Table 2* provides the MAD
of both the velocities and the water levels for each applied Manning coefficient ($n$). Lower values of MAD represent more
optimal results.

Table 2 Mean Average Deviation for Roughness optimisation

| Manning coefficient [$s/m^{1/3}$] | MAD of Current metre [m/s] | [%] | MAD of LSPIV [m/s] | [%] | MAD of water level [m] |
|---|---|---|---|---|---|
| 0.012 | 0.104 | 8.2 | 0.097 | 9.2 | 0.095 |
| 0.013 | 0.11 | 8.7 | 0.077 | 7.3 | 0.067 |
| 0.014 | 0.124 | 9.8 | 0.069 | 6.7 | 0.063 |
| 0.015 | 0.144 | 11.3 | 0.067 | 6.4 | 0.075 |
| 0.016 | 0.162 | 12.8 | 0.071 | 6.8 | 0.099 |
| 0.017 | 0.176 | 13.9 | 0.075 | 7.1 | 0.145 |
| 0.018 | 0.196 | 15.4 | 0.085 | 8.1 | 0.193 |

The first model simulation which was set at 0.018 $s/m^{-/13}$ shows a relatively high average deviation (LSPIV: 15.4 % & CM:
8.1%) of the surface flow velocity and an overestimation of the water level by 19.3 cm. This results in a substantial widening
of the river due to the uniform 'flat' floodplain. Both the velocity and the water level indicate a better performance when lower
roughness value are applied since less resistance means faster flowing water and a lower water level with equal discharge.
After further reductions in roughness values, results indicate that velocity and water levels are optimal when the Manning is
set at either 0.013 $s/m^{1/3}$ or 0.014 $s/m^{1/3}$. Since the CM measurements had to be performed from a boat, we expect higher





uncertainties in these measurements. Hence, 0.014 s/m$^{1/3}$ (highlighted in grey in Table1), is selected as the optimal roughness coefficient of the main channel


The model validation was performed based on a visual analysis of the alignment between the measured RTK tie line and the simulated water level. *Figure 8* shows the RTK tie line which was measured along the water line and the simulated flow at $Q$ = 191 m$^3$/s, $n$ = 0.014 s/m$^{1/3}$ (main channel) and $n$ = 0.040 s/m$^{1/3}$ (floodplain). In the absence of varying seasonal gauge readings, the alignment between the RTK tie line and the simulated water line on the right bank of the river provides visual evidence of

good model performance.

After the model was setup and evaluated, simulations ranging from 5 m$^3$/s to 3,000 m$^3$/s with increments 100 m$^3$/s of were performed. *Figure 9* presents four rating curves derived from D3DFM; one based on a single channel Manning coefficient

(derived from dry-season flow survey in the main channel), the second is based on a combination of 2 coefficients (main channel and floodplain), the third curve shows the rating curve based on a 1D HEC-RAS model and the final curve is based on the conventional gauging method from WARMA. The discharge measurements are visualised in relation to a 95% confidence interval of the WARMA rating curve. In addition to the confidence interval, we evaluated the significant differences among the curves based $E_{ns}$ and *Pbias* in relation to the WARMA curve.


The D3DFM based model which combines two different roughness coefficients more closely resembles the WARMA curve than the 1D HEC-RAS curve and the D3DFM which applies only one roughness for the entire terrain. This is particularly the case for high flow conditions. This result may be attributed to better optimisation of the roughness coefficients (compared to 1D or 3D with only one Manning roughness) which acknowledges the fact that roughness in the main channel is different from

roughness in the floodplain.  It must however be noted that comparing with the relationships of WARMA and 1D HEC-RAS is only insightful to a certain extent as the experiment was not conducted at the exact same location as where the WARMA rating curve is maintained. Possible differences in the river geometry may cause that our results are not entirely equivalent with WARMA's rating curve. The final stage-discharge relationship is expressed by *figure 10* and e*quation 3*. This relationship should function as a basis on which adjustments can be made based on newly available stage-discharge data. Note that the

river geometry will most likely change over time, due to the sandy bed-level, and therefore the constants are not stable over time.

$$Q = 3.42[h - h_0]^{3.39} \hspace{4cm} Eq\ 3$$





## 4.2. Comparison of discharge based on varying GCP numbers.

To assess the impact of the number of ground control points on the bathymetric chart and therewith on the modelled discharge,
charts created with different GCP numbers were used to run the same hydraulic model with similar boundary conditions.
*Figure 11* presents the rating curves of all four distributions.

Assuming the bathymetry based on 17 GCPs as the control, we plotted a 95% confidence interval on its rating curve. The
confidence interval was plotted based on OLS regression results. These results are presented in *Annex 1*. The $P_{bias}$ and $E_{ns}$
results indicate very similar curves derived among bathymetries based on 5, 9, 13 GCPs; PBIAS [3%, 0.7% & 0.6%] and NSE
[0.982, 0.998, & 0.999] respectively. All 4 curves fell within the 95 % confidence interval of the control curve (17 GCPs). It
must be noted that the bathymetry up until 191m$^3$/s is determined by the ADCP/RTK measurements and therefore the number
of GCPs does not influence the curve up until this point. In this study, a minimum of 5 GCPs spread over 25 ha is sufficient
for accurate discharge estimation. We draw a conclusion that for the purposes of physics based river rating, a ratio of 5 ha/GCP
is sufficient to accurately estimate discharge. However, in all instances including terrains less than 1 ha, the base-
level/minimum number of GCPs required is 3 to allow for triangulation (Oniga et al., 2020). Finally, it is important to note
that the distribution of the GCPs is likely to influence the final chart drastically as the most uncertain areas will be at the
borders of the bathymetry (mostly due to the bowling effect). Therefore an optimal GCP distribution will not only be
representative of the full spectrum of elevations, but, priorities placement of GCPs on the edges of the terrain being mapped.


## 4.3 The impact of uncertainty in proxies of flow on discharge estimation.

Finally, we investigated the impact of proxy uncertainties (river width and water level) on discharge estimation. With proxies
we mean here variables that can be more easily observed operationally. We imposed uncertainty based on the resolution of
satellite sensors we may rely on such as IceSAT-2 for river depth and Sentinel-1/2 for river width. *Figure 12* presents the
relationship between discharge and river width. The graph also highlights 2 different potential error intervals, +/- 5 meters (90
%) and +/- 10 meters (95 %) so as to visualise the amount of uncertainty which corresponds with specific sections of the
terrain.
If river widths would be used, this would results in high levels of flow uncertainty below 150 meters. These higher levels of
uncertainty are as a result of low width sensitivity to changes in flow below 150 m. The low sensitivity in this low flow stage
can be attributed to the steep bank, i.e. as flow increases the depth rises quickly but there are minimal changes in width. During
medium level flows, between 150 m. and 370 m., results indicate lower levels of width uncertainty i.e. high river width
sensitivity.  The high sensitivity in this medium flow stage may be attributed to the gentle sloping floodplain (more stable
roughness coefficient), i.e., as flow increases the width rises significantly faster than the water level. Finally, higher levels of

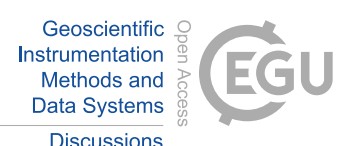

width uncertainty are noted during high flows (above 370 meters). This region experiences low width sensitivity to changes in flow. The causal factor is inundation of entire floodplain, which has not been schematized in the hydraulic schematization.

Similar to width, water level uncertainties also result in varying discharge estimates. *Figure 13* presents the relationship between discharge and water level as simulated by D3DFM. The graph also highlights 2 different potential error intervals, +/-

10 cm (90 %) and +/- 20 cm (95 %). These error intervals assist us in visualisation of the amount of uncertainty in flow that can be expected from using water levels as proxy. . For lower flows (<1 000 m$^3$/s), results indicate lower levels of water level uncertainty i.e. high water level sensitivity. The justification for the high sensitivity in this low flow stage can be attributed to the steep bank, i.e. as flow increases the depth rises quickly but there are minimal changes in width. During medium level flows, between 1 000 m$^3$/s and 1 500 m$^3$/s, results indicate higher levels of water uncertainty i.e. low water level width

sensitivity.  The low sensitivity in this medium flow stage may be attributed to the gentle sloping floodplain, i.e. as flow changes, the water level does not change significantly. Finally, during high flows the floodplain is inundated with water, thus, the expectation is that in this regime high water level sensitivity i.e. low water level uncertainty. Contrary to our expectation, this segment experiences high water level uncertainty. This may be because the magnitude larger or because of lateral flow of water below thick forest on the left bank and disturbances from unnatural infrastructural development (e.g. the road) on right

bank maintains high levels of uncertainty.

As shown in figure 10 and 11, the proxies of flow (water level and river width), are antagonistic in nature. This implies that when one of the proxies exhibits high uncertainty, the other is more likely to presents low levels of uncertainty.

We note that different proxies of flow, namely water level and river width, perform optimally at different segments. At low

flows the shape of the wet river channel (steep slope) is more likely to induce high water level sensitivity and low river width sensitivity to changes in discharge. At higher flow levels the shape of the wet river channel (gentle slope) is more likely to induce low water level sensitivity and high river width sensitivity to changes in flow. At even higher flows, ideally, the floodplain is inundated and becomes insensitive to river width. In the absence of more accurate discharge estimation methods, the water level is once again the more reliable proxy. Above the natural levee, the assumptions of the schematization of the

D3DFM model no longer hold, and therefore any flows above that level should not be considered reliable.

## 5 Conclusion and Recommendations

The study reaffirms and provides insight into the potential of applying low-cost and readily available technologies for river monitoring. The methods described in the study are well within reach of water authorities with limited resources and are

particularly useful for small to medium sized rivers in sub-Saharan Africa. The D3DFM discharge model resembles actual





river in depth, width and location when using a combination of two Manning's coefficients (0.014 s/m$^{1/3}$ & 0.040 s/m$^{1/3}$) and a discharge value of 191 m$^3$/s.

Based on the $P_{BIAS}$ and $E_{ns}$ values, there is no significant difference in estimated discharge for bathymetries reconstructed based on 5, 9, 13 and 17 GCPs. 5GCPs are sufficient to simulate a curve which falls within the 95% confidence interval of a WARMA

curve (control). Therefore, 5 GCPs are adequate for physically based river rating on condition that the GCPs are accurately measured using an RTK GNSS and are optimally distributed to represent the full spectrum of terrain elevations.

The slope, which is an important input to the model, must be measured as accurately as possible for the longest possible distance along the water line. Ideally, measuring the waterline height at 200 m intervals for a 5 km stretch is sufficient to avoid the impact of wave distortions. The impact of backwater distortions is of particular concern for high water levels as opposed

to low water levels and therefore a longer measuring distance is required in high water level instances. However, the magnitude of slope has a bearing on the length that is required to reduce the impact of backwater distortions, i.e. in Luangwa's case, a long distance would be needed but for streams with a large bottom slope, a much shorter distance is sufficient. Furthermore the stretch chosen for observation must be long enough to cancel out the effects of sand banks (uneven silt deposition) which may have an impact on the slope accuracy. However, identifying and measuring such long stretches is problematic due to

difficult terrains and inaccuracies caused by the need to move the base station. The most feasible compromise is to use one base station location and then measure continuously for as far as possible to both sides, use correction via satellites, or use a spirit level. In that way the relative accuracy stays the same and will be very good.

We determined that the proxies of flow (water level and river width) perform well at different stages of discharge. For instance,

at low discharge values and steep banks, the water level is more sensitive to changes in flow, thus more accurate. For higher discharge values and gentle floodplain slopes where the floodplain fills up, the river width is more sensitive to flow changes and thus more appropriate to use. As a result of the two proxies acting antagonistically in performance, a combination of both methods in different flow regimes gives a more accurate flow monitoring assessment. Alternatively, determining the river geometry and then deciding on which proxy would be most helpful to measure i.e., for gently sloping riverbed using the width

since a slight change in discharge will have a larger impact on the width and therefore be easier to measure. And vice versa for steeply sloping river beds (rectangular channel will be only interesting for water level measurements).

We reiterate that the accurate measurement of a tie line is critical not only to correct the doming effect, but to provide an extra validation check for the hydraulic model. In this study we demonstrated that this is feasible and affordable using a simple combination of an RTK GNSS and a mobile cart. The tie line must be measured simultaneously with the river discharge so

that it can be compared against the simulated water line as derived by the hydraulic model. Finally, we recommend that the approach is applied in the dry season so as to minimize the amount of water flowing in the river for more efficient photogrammetry processing. However, it is important to occasionally measure flows and corresponding water levels at different times of the year so as to validate the efficiency of the model simulation and differentiate roughness in the main channel and floodplain.




# A

# 1 D HEC-RAS model

In this annex, we describe a preliminary study which was conducted in order to determine the optimal roughness coefficient during high flows. The preliminary research was conducted in close proximity to the study currently in question. Both study locations have similar geophysical and hydraulic properties, thus, are comparable. The research methodology was divided in four stages. The first stage was data collection of discharge, bathymetry and aerial data. A DJI phantom 4 Unmanned Aerial Vehicle (UAV) with a 12 MP camera was used to collect. The second stage was processing of images and transects collected using the Unmanned Aerial Vehicle (UAV) and Acoustic Doppler Current Profiler (ADCP) respectively. The images were merged together and used to reconstruct the dry topography through photogrammetry. The third stage involved hydraulic modelling using the HEC-RAS model. The 1D steady-state hydraulic model was built and calibrated based on the ADCP measurements. In the final stage, the more physically based rating curve from the hydraulic model was compared with a traditional rating curve from the Zambian Water Resources Management Authority (WARMA).

The model output was evaluated by the Root Mean Squared Error (RMSE). The lowest value for the RMSE is obtained for a Manning's roughness coefficient of n = 0.040 s/m$^{-1/3}$. According to literature this seems to be a reasonable value. We proceed to utilise this roughness value in the current study as a representation of the optimal roughness during high flows.

.




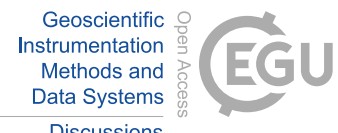

# B

# OLS Regression Results


```
                        OLS Regression Results
==============================================================================
Dep. Variable:                 log_Q   R-squared:                       0.995
Model:                           OLS   Adj. R-squared:                  0.994
Method:                Least Squares   F-statistic:                     2976.
Date:               Thu, 03 Nov 2022   Prob (F-statistic):           1.32e-19
Time:                       07:58:12   Log-Likelihood:                 28.325
No. Observations:                 18   AIC:                            -52.65
Df Residuals:                     16   BIC:                            -50.87
Df Model:                          1
Covariance Type:           nonrobust
==============================================================================
                 coef    std err          t      P>|t|      [0.025      0.975]
------------------------------------------------------------------------------
Intercept      1.0797      0.030     36.337      0.000       1.017       1.143
log_control    2.8092      0.051     54.552      0.000       2.700       2.918
==============================================================================
Omnibus:                       16.412   Durbin-Watson:                   0.683
Prob(Omnibus):                  0.000   Jarque-Bera (JB):               16.859
Skew:                          -1.571   Prob(JB):                     0.000218
Kurtosis:                       6.551   Cond. No.                         5.28
==============================================================================
```

**Annex 1  OLS regression for Control**





## 6 Data Availability

Images used to carry out the ODM photogrammetry can be found on https://doi.org/10.4121/21557148.v1

## 7 Author Contributions

Hubert Samboko performed conceptualisation, data curation, formal analysis, investigation and writing the original draft. Hessel Winsemius performed conceptualisation, reviewing, editing and supervision. Sten Schurer performed data collection, curation and investigation. Hodson Makurira performed supervision, reviewing and editing. Kawawa Banda performed

reviewing and editing. Hubert Savenije performed funds acquisition, supervision, reviewing and editing.

## 8 Competing interests

The authors declare that they have no conflict of interest

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

**10 Acknowledgments**


This work is part of the research programme ZAMSECUR with project number W 07.303.102, which is financed by the Netherlands Organisation for Scientific Research (NWO). This research received and continues to receive support from the University of Zambia and the Zambian Water Resources Management Authority.

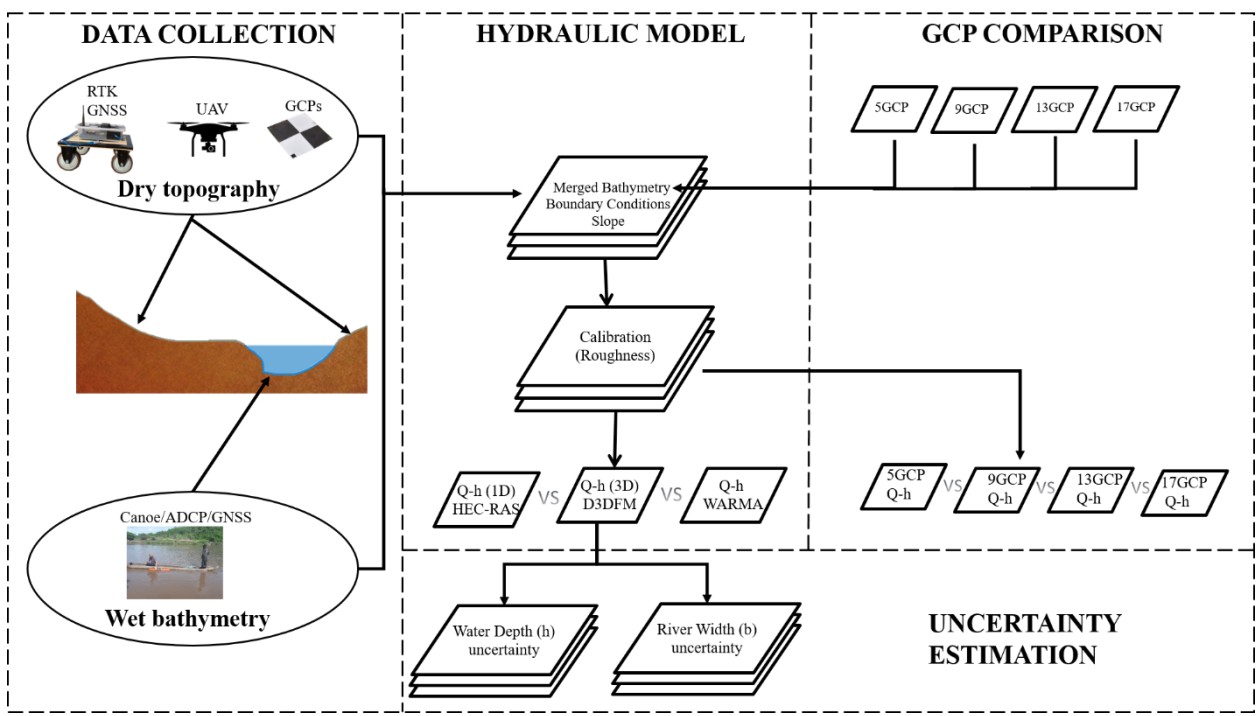

**Figure 1 Schematic of experimental procedure**





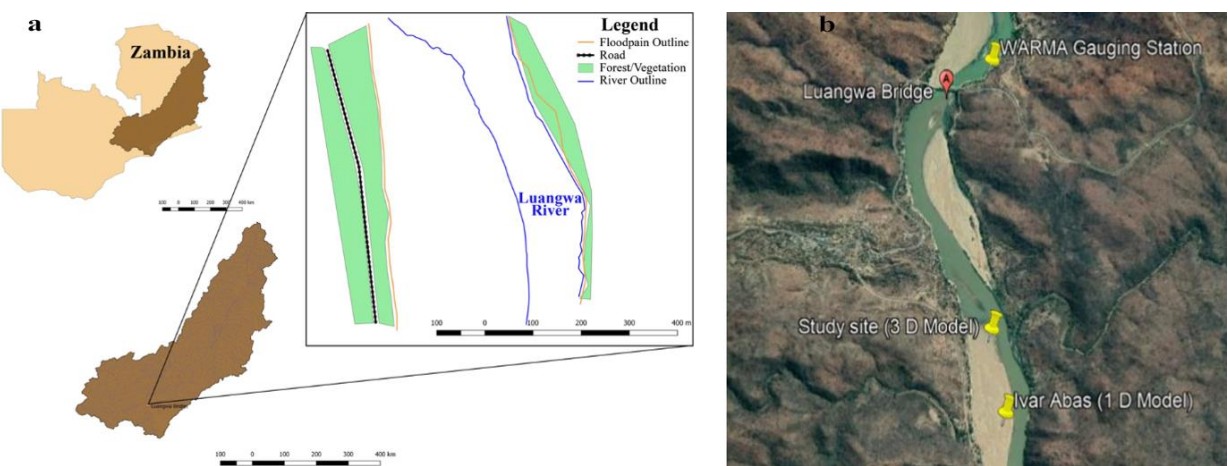

**Figure 2 (a) Study site along Luangwa River (b) location of study site in relation to other comparison sites (Google Maps, 2022).**

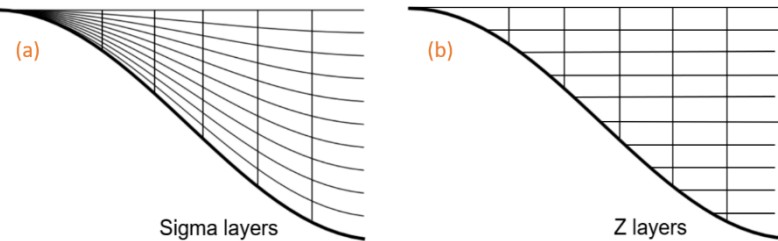

**Figure 3 Representation of (a) Sigma and (b) Z layering methods in D3DFM (Deltares, 2020)**

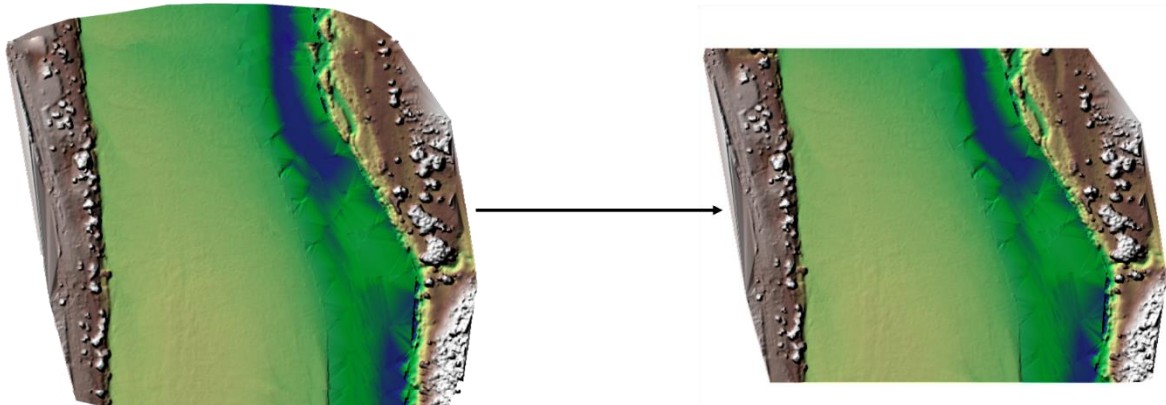

**Figure 4 DEM which has been volumised and cut on both sides**



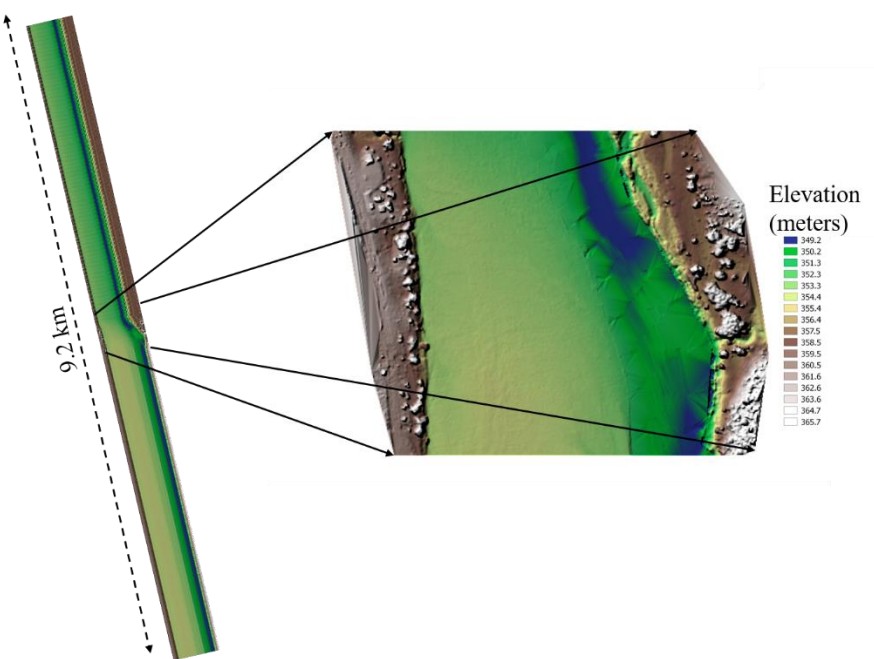

**Figure 5 Elongated elevation model imported into D3DFM**

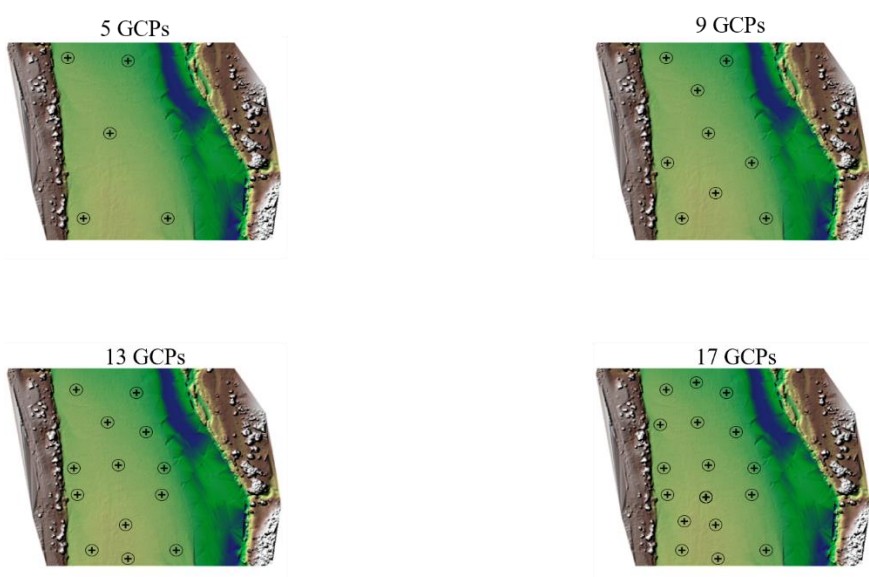


**Figure 6 GCP distribution along floodplain of the Luangwa River**



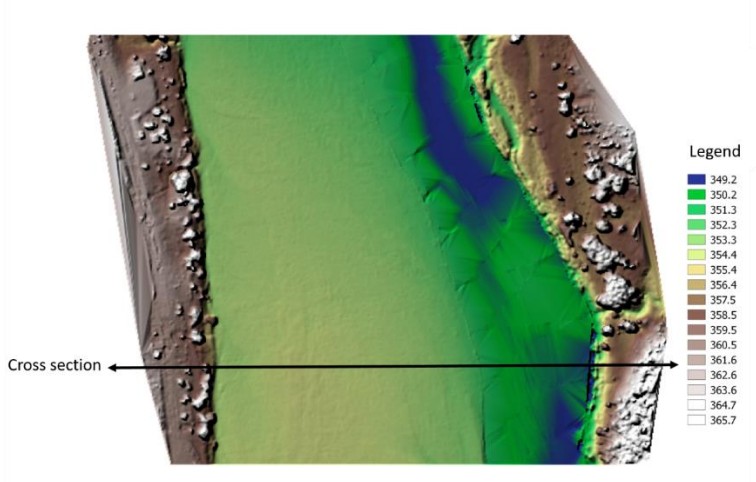

**Figure 7 Location and orientation of cross-section**


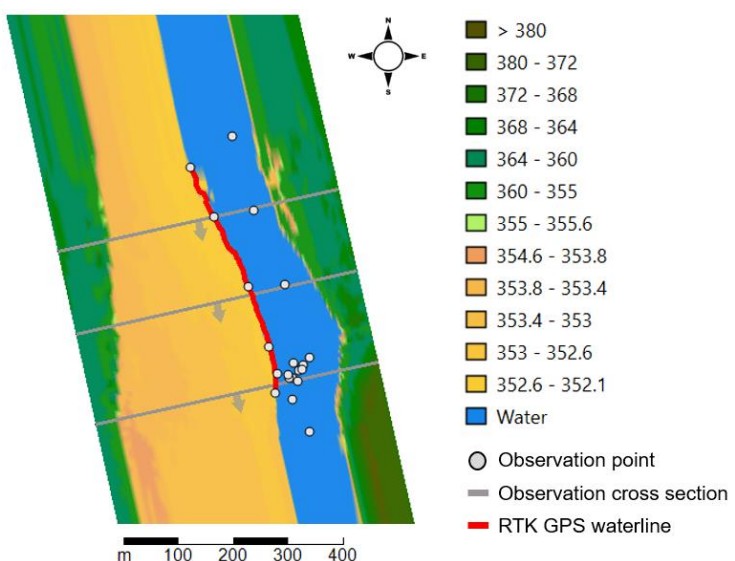

**Figure 8 Visual representation of the discharge model at a discharge of 191 m³/s with n = 0.014 s/m$^{1/3}$**





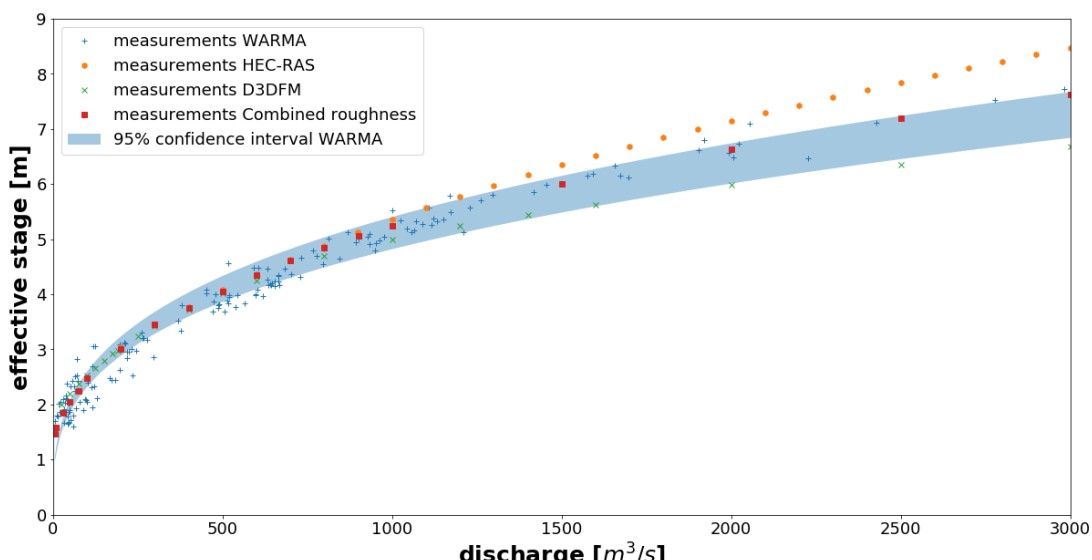

**Figure 9 Rating curves comparing D3DFM with convetional methods**

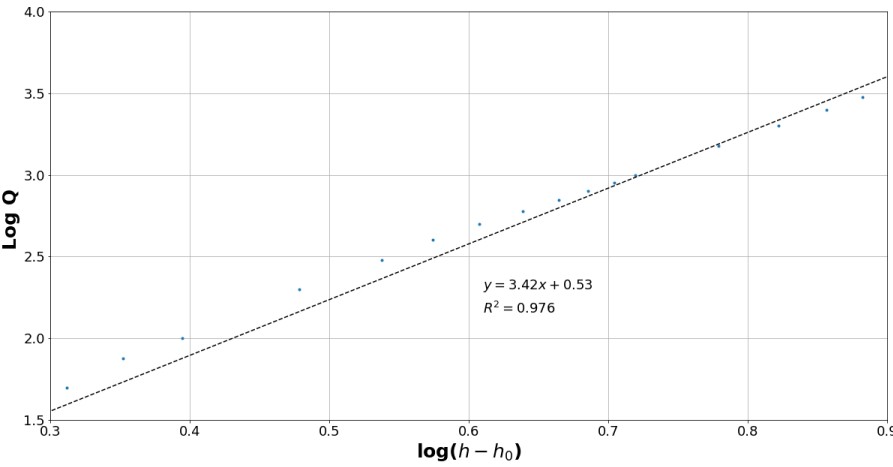


**Figure 10 (Logarithm) Discharge vs stage relationship: combined roughness**





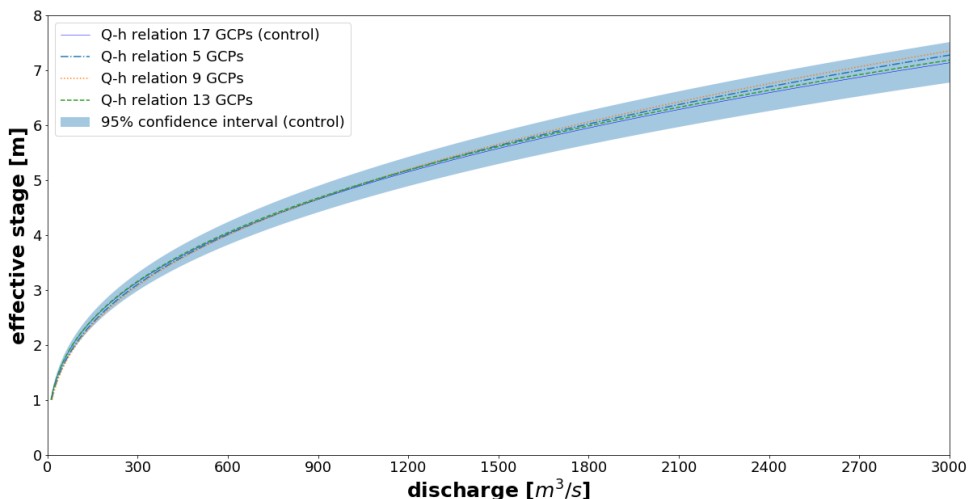

**Figure 11 Comparison of Rating curves generated based on varying GCPs**

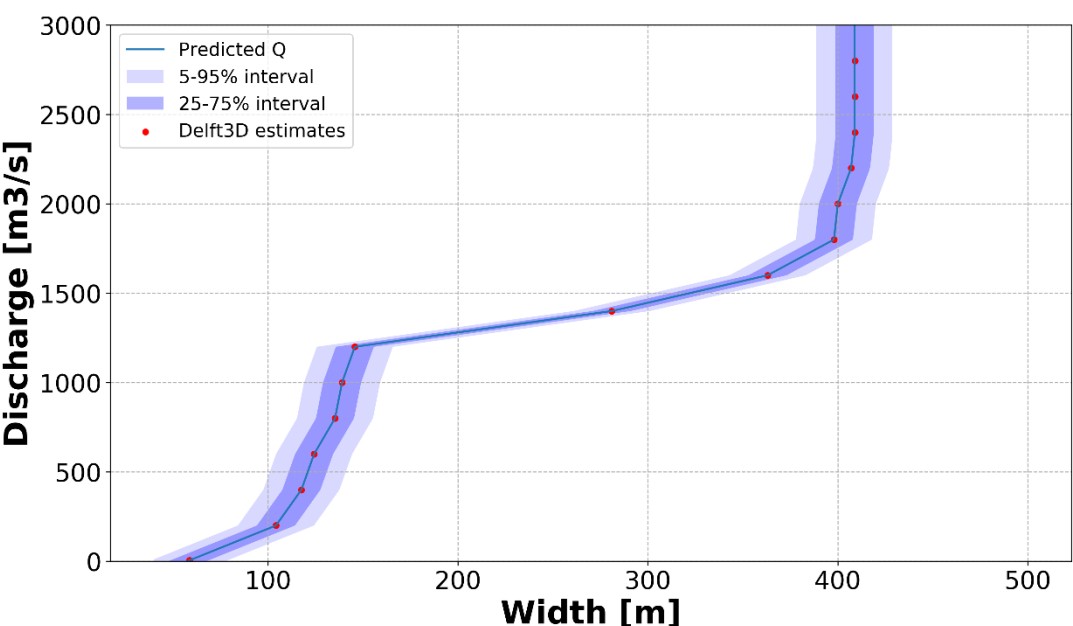

**Figure 12 Discharge vs width relationship**





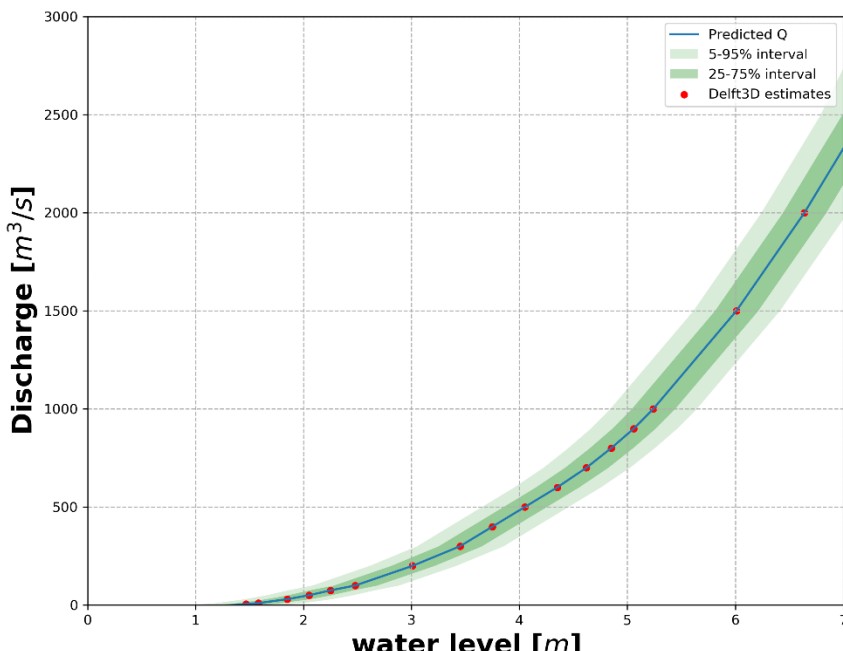

**Figure 13  Discharge vs water level relationship**