# Peer review of "Towards Affordable 3D Physics-Based River Flow Rating: Application Over Luangwa River Basin"

_Geoscientific Instrumentation, Methods and Data Systems, 2022_

## Author Response (AR1)

**Reviewer comment**

In submission of Manuscript titled:

**Towards Affordable 3D Physics-Based River Flow Rating: Application Over Luangwa River Basin (Hubert Samboko)**

On behalf of the authors of this manuscript allow me to express my utmost gratitude for taking the time to read through our manuscript. The comments you have provided are eloquently articulated. We are confident that implementation of the suggested recommendations/corrections will improve the quality of our work significantly.

With respect to the first major comment (Use of one rating curve for a channel …………) we acknowledge that the use of more than one rating curve would help assess the robustness of the 3D model. To that end we have rephrased our problem statement to concentrate more on the fact that traditional methods are mostly based on point data. We therefore proceed with the study with the aim to assess if the UAV system provides better accuracy due to the higher resolution.

With respect to the second major comment (Superiority of 3D model compared with 1D model) we have introduced a discussions section at the end of the results. Below are specific comments and have made the changes as outlined below.

| Comment | Response | Changes to Manuscript |
|---|---|---|
| L15: suggest adding 'multi-beam' before echo sounders (if you want to include single beam one, use parenthesis) | Suggestion noted and applied. | Added 'multi beam' as suggested |
| L17: hardware(s)? | Suggestion noted. The word hardware was notably not appropriate and slightly confusing. | Replaced the word hardware with a more appropriate word 'system' |
| L24: 'determine how .... methods' incomplete sentence? | Suggestion noted | |
| L25: 'the' hydraulic model? | Suggestion noted and applied. | The word 'the' has been added |
| L29: Meaning of 'physics-based' should be defined in the abstract if use it in the abstract. (Of course, we can guess that author is thinking 3D flow modelling is physics-based, but 1D flow model can be said physics-bases since the shallow-water | As opposed to defining the term physics-based in the abstract we opt to substitute word with the more specific reference to 3d Hydraulic model to make the sentence easier to understand for the reader | Swapped the word physics-based with 3D hydraulic model |

| | | |
|---|---|---|
| equations are based on the Navier-Stokes equations which is used for 3D flow model.) | | |
| L30: permanent -> stable or immobile (or some others?) ('permanent' sounds something like (very rigid) bedrock but the target site seems sand bar) | Suggestion noted and applied. | Replaced the word permanent with stable |
| L34: 'is most promising to use' A bit vague and logical flow of the sentence is not clear. | Suggestion noted. | The words ''is most promising to us' have been replaced by 'more accurate' |
| L36: remove 'b' before 'hydraulic' | Suggestion noted and applied. | The letter has been removed |
| L44: 'implement' would come prior to 'validate'? | Suggestion noted and applied. | Order rearranged |
| L45: flow rate would be one of the most important inputs for the flow model (maybe authors see the flow rate as output, but such standpoint is not explained yet). | We agree that the flow rate discharge is one of the most important inputs) | We have now rephrased the statement to acknowledge the importance of flowrate as an input but to also acknowledge our focus on geometry. L48 'Assuming that the flow rate is constant….' |
| L59: 'It is within this technological gap....' hard to read? | Suggestion noted | Sentence has altered to make it more easily read |
| L65: 'The process of applying' -> 'Distributing and surveying'? | Suggestion noted and applied. | Process of supplying has been replaced by the suggested ''Distributing and surveying' |
| L81: The sentence seems incomplete? | The sentence was indeed incomplete and has been edited to make it easier for readers to understand | Research question has been edited to better describe the question. And the questions have been numbered |
| L91: 'within' ->'for' (not confident | Suggestion noted and applied | 'within' has been replaced with 'for' as suggested |
| L92: 'a number of times until' -> 'with different flow rates'? | Suggestion noted and applied | ' a number of times' replaced with 'with different flow rates as suggested |

| | | |
|---|---|---|
| L99: Combining DEM (obtained by LiDAR or photogrammetrically) and bathymetry (obtained by echo-sounding) are quite common in river engineering, so maybe introducing not only authors' output but other works would be good. | We have noted the comment and added another reference | Similar study by Alvarez (2018) as been introduced to provide more reference to the concept of combining DEMs |
| L122, 'the 2 other sites' a bit vague? 'the two sites discussed in previous works (one or two citations)'? | Suggestion noted and statement has been added to explain the other 2 sites in more detail. | Explanation on which particular sites are being referred to has been added |
| L228: add 'software' before 'D-Flow Flexible...'? | Suggestion noted and applied | The term software has been added before D flow |
| L132: 'in thickness' -> 'their thickness'? | Suggestion noted and applied | In replaced with their as suggested |
| 162: 'point cloud' suddenly appeared. | Suggestion noted | We add the term point-cloud as a product of 'volumizing. This assists in the introduction in the stated comment. |
| L164: 'does not affect the water levels' is it realistic? (For subcritical, non-uniform, varied flow, local water level is affected by the downstream flow, I think. Of course, I understand what you want to say, but you can say it with different expressions) | Suggestion noted and applied | We have tried to rephrase the sentence to clearly state that the backwater effect is the phenomena we attempt to manage |
| L164: 'A small selection ... is taken' not clear. | | We rephrase the sentence to improve readability |
| L 186: 'the coordinates of known surface velocities' -> ' the surface velocity distribution'? | Suggestion noted and applied | Coordinates of known surface velocities'Replaced with surface velocity distribution' as suggested |

| | | |
|---|---|---|
| L 187: 'the coordinates of known water levels' -> ' the water level profile'? | Suggestion noted and applied | coordinates of known water levels' Replaced with water level profile' as suggested |
| L 190: better to show the equation of MAD for improve clarity (it's not strong suggestion but maybe help some readers to understand) | Suggestion noted and applied | Equation has been added to help with readers understand the MAD equation |
| L202: 'iterations which estimated the water level based on slope' a bit vague, and better to explain more details. I think we can have results quite similar to the result of WARMA if the 'expert' do the iteration? | We have added details to improve readability for readers | We have added few details as per how the downstream water levels can be determined |
| L222: Is this a sub-section title? | Suggestion noted and applied | This is a subsection and has now been correctly highlighted |
| L232: '=/-' -> '+/\'? | Suggestion noted and applied | Signs corrected as suggested |
| Does the table necessary? (Surface velocity seems to be used for both calibration and validation, it's good if there is some discussion about it) | We feel that the table is necessary and have added descriptions which define that there was enough data to distribute among calibration and validation | Table edited to improve explanation of the use if surface velocity for both calibration and validation |
| Table 2: How were the distribution and sample size of three properties discussed in the table? (Result of current metre shows no minimum. Is this show the problem of the current metre survey? (it's hard comment so able to skip)) | In our opinion the current meter results ae not extremely reliable. We suggest that this is caused by the unstable measuring conditions i.e. the swaying canoe. | The sample size was stated on L199 'Note that this score is based on 5 …..' |

| | | |
|---|---|---|
| L261: A bit difficult to understand how this conclusion can be obtained from table 2. (LSPIV shows the minimum with 0.015 s/m^(1/3) but why choose 0.013 as a conclusion) | We agree that the two main factors should be LSPIV and water levels. This leaves two values as the options. The minimum Water Level MAD which corresponds with 0.014 and minimum LSPIV MAD corresponds with 0.015. Applying the current meter MAD as the tie breaker we lean towards a similar conclusion that 0.014 produces the most accurate results | We have rewritten the statement in line with the logic presented by the reviewer. We conclude that the roughness is optimal at 0.014. |
| L273: remove 'of' at '100 m3/s of were'? | Suggestion noted and applied | The word 'of' has been removed |
| L274: 'four rating curves derived from D4DFM; one based on....' Misleading? (two rating curves were based on D4DFM but two others were not?) | We note that the statement was misleading and have corrected as advised | Statement corrected to say there are four rating curves. As opposed to four rating curves based on D3DFM |
| L299: Spell out 'OLS'? | Suggestion noted and applied | The full term Ordinary Least Squares (OLS) has been spelt out |
| L299: P_{bias} and E_{ns}: compared with 17GCPs result or WARMA? (maybe with 17GCPs but better to indicate( | Suggestion noted and clarification has been made. | We have clarified in text that the comparison is with the 17GCPs to avoid confusion. |
| L314: Better to indicate the reference if the uncertainty used here (also for L329) | Suggestion noted and applied | We have added the reference (Coppo Frias (2023) and (Filippucci et al., 2022)) |

| | | |
|---|---|---|
| L323: 'more stable roughness coefficient' a bit unclear. | We have decided to remove the comment which was unclear. | The statement has been removed |
| L326: 'schematized' and 'schematization' are used very close, maybe can be rephrase to improve readability. | Suggestion noted and applied | We have rephrased the sentence so as to avoid repletion of the word schematize. |
| L342: figure(s)? | Suggestion noted and applied | We have added the letter 's' |
| Bibliography of Kim. Y (2006) could be edited. | Suggestion noted and applied | We have corrected the error in the reference for Kim Y |
| Figure 9: Labels 'measurements HEC-RAS, D3DFM, Combined roughness' are a bit confusing. something like' estimate with HEC-RAS, D3DFM (single roughness), D4DFM (combined roughness)'? | Suggestion noted and applied | We have renamed the graph names according to the suggested names to improve readability |
| Figure 10: The regression line seems weighted to high flow. Is there any reason? (based on annex B, it seems logs are applied to both axes) | There is no particular reason to weight to high flow. The results were | |

**Reviewer 2**

Firstly, we would like to sincerely thank the reviewer for the meticulous attention to detail especially in regards to the papers potential contribution to science. The reviewer provides useful comments and recommendations which we believe will significantly improve the manuscript if implemented adequately. We acknowledge the shortcomings that have been identified in terms of comparison between the UAV system and the traditional

estimation methods. As suggested, we will refocus the problem statement showing scientific evidence of how point measurements fail to estimate river discharge. In general the reviewer points us in the right direction with respect to the need to add more discussion of results (1D vs 3D model, reliability of results, number of rating curves required). We have gone through all 24 specific comments and have made the changes as outlined below.

| Comment | Response | Changes to Manuscript |
|---|---|---|
| L17 – What do you mean with "hardware"? Would it not be better to use the word "system" Also, the sentence "In short, the hardware can be used to produce the geometry" is confusing. Do you mean river geometry? The sentence, as it is, seeming to be incomplete or somehow needs to be related to the previous sentence or the following one. | Suggestion noted and applied. | We have replaced hardware with apply system |
| L22- I recommend mentioning the novelty/contribution in the abstract. This can be place before objectives and after briefly explaining the problem. | Suggestion noted and applied. | We have added the novelty to the abstract 'Traditional methods of river monitoring are based on point measurem………..'  Line 18 |
| L24 – Instead of using semicolon, I would recommend alphabetic numerating of the objectives (a, b, c, etc). This will allow the reader to easily differentiate between them. | Suggestion noted and applied. | We have changed from the use of semi colons to alphabetic numeration to improve readability |
| L32 –Using the number 9 in parentheses is confusing, it only makes sense when reading the methods in the paper. I recommend removing it and leaving the sentence "beyond an optimal number" or change it to | Suggestion noted and applied. | We have removed the number 9 |

| | | |
|---|---|---|
| "beyond an optimal threshold of 9 GCPs". | | |
| L36 – remove "d". | Suggestion noted and applied. | We have removed the letter d |
| L44 – I would use the word "estimation" rather than "monitoring". Monitoring can be confused with sensing, then it is better to clarify that models are useful tools for prediction rather than monitoring/sensing. | Suggestion noted and applied. | We have replaced monitoring with apply estimation |
| L45 – Use the word "apply" rather than "implement". | Suggestion noted and applied. | We have replaced implement with apply |
| L75 – I would remove the word "robust". It could be argued that more measurements and rating curves are needed to make the method robust. I leave it for your consideration. | Suggestion noted and applied. | We have removed the word robust |
| L80 – Research questions are objectives rewritten as questions. Although, there is nothing wrong with this, it is repeated information. If the authors want to leave the research questions, I suggest modifying them. I leave it for your consideration. | Suggestion noted and applied. | We have attempted to edit the questions to make them more easily understoodand also different to objectives |
| L86 to L95 – It is not easy to follow the steps as they are | Suggestion noted and applied. | We have numerated the text |

| | | |
|---|---|---|
| written. I suggest numerating them (i, ii, etc). | | |
| L119 - Were the measurements of flow and water level contemporary with those of GCP and bathymetry? If so, what year was it (2022)? Could you please add in a table the data collection date, or maybe add this in table 1. | done | We have stated in text that the data was contemporary (collected at the same time) We have also edited the table. |
| L210 – Add name of variables (O = observation, P, x, etc) | Suggestion noted and applied. | The representations have now been labelled |
| L222 – Seems incomplete | This seemed incomplete because it was a heading which had been omitted when highlighting | The section has now been correctly labelled and highlighted as a subsection |
| L237 – Results should be section "3". Previous section is "2 Material and Methods". | Suggestion noted and applied. | Correction of the section numbering has been made |
| L312 – I don't recall you mentioning satellite data in the "2 Material and Methods" section. You need to add this in section "2.7"? | Suggestion noted and applied. | A note of the satellite data and its implication of uncertainty was indeed missing and as subsequently been added to section 2.7. |
| L353 -Where is the "discussion" section? I consider it very important to discuss the differences of using a 3D model vs the 1D model. Also, the use of more than one rating curve (if using only one discuss why?). The advantages of your method over others, etc | Suggestion noted and applied | A discussion section has been added as suggested to describe difference between 1 D and 3D model among other discussions |
| Figure 2 and Figure 5 - Legend and scales need to be bigger. It is difficult to read. | Suggestion noted and applied | Scales and legends have been increased in size |

| | | |
|---|---|---|
| Figure 3 – I don't think this figure provides important information. I would remove it also because it does not follow the same format as other figures. | Suggestion noted and applied | The figure has been removed |
| Figure 4. – Subfigures require identification (a & b) and a respective legend. Also, they look identical to me. Make evident the "volumised and cut on both sides". What do the colours mean? -depth (m)? Add a colour bar. | Suggestion noted and has subsequently been merged to figure 5 as suggested since they are similar. | Figure 4 and 5 have been merged to one. Volumised part has been made clear |
| Figure 5 – I think figure 4 and figure 5 can be a single one (a, b, and c). | Suggestion noted and applied | Figure 4 and 5 have been merged to one |
| Figure 6- Use identification for subfigures (a, b, c, etc) and add their respective legend (5 GCP, 9 GCP, etc). | Suggestion noted and applied | Identification for subfigures has been added and legend has been added |
| Figure 7 – Bigger legend. | Suggestion noted and applied | Legend size has been ncreased |
| Figure 8 – This is a good example of a figure. | Noted with thanks | |
| In general, in the text there is a discrepancy in the format. Sometimes you use a space between new lines (L135, L160, L180, etc), sometimes you don't (e.g., L50, L59, L114, etc). Also, | Suggestion noted and applied | We have made an effort to correct all these errors and discrepancy's across the entire document and hope that it is now up to standard |

| there are tabs where they shouldn't be (L221 and L214). In terms of the figures, you use different colour and letter sizes (e.g., Figure 1 a, b labels in black vs figure 2 a, b labels in parentheses and in orange). | | |
|---|---|---|

---

## Referee Report (RR1)

The research shows how state-of-the-art technology can be applied to estimate river flow at a small spatial scale. The authors took into consideration the comments made previously and significantly improved the contribution. The methods and results were clearly explained. I consider that this research has substantial scientific merits for its acceptance. The manuscript requires minor corrections specially in formatting, see below for my comments.

L38 – I do not think that your study has applied low-cost technology to river monitoring. In-situ monitoring is cheaper. However, I understand that compared to other systems (for example, a plane with Lidar), yours is more economic. I think it would be good to clarify this in the sentence. Same in L361)

L40 – Here you talk about the validation process, but it is not very clear what it includes. It is worth adding a sentence explaining the methods (e.g., how calibration and validation were done).

L41- While the specific objectives are important to understanding the research, I do not think it is worth mentioning them in the abstract. It would be better to mention your overall aim. This will also give space to include the methods to understand the summary of the results.

L107 – Please provide a summary of the method of Samboko et al. (2022) and then use the reference. If this is related to L110 please make it clear. Although it looks like L110 is related to Alvarez (2018).

L136 – Use of a single set of parentheses "(D3DFM, Deltares, 2020)".

L157 – I don't see the point of using a subsection (also make it as "2.4.1" if you are including it).

L160 – Same comment on Samboko et al. 2022. Include a summary here or above (L107).

L214 – Removed the extra dot before referring to Moriasi et al. (1983)

L219 – This sentence should not be in bold.

L220 – Tabulation. Maybe this is related to document conversion, please check final draft of the PDF before submitting.

L217 – Tabulation errors (same as previous comment).

L266 – Superscript error "-/13". Also use for the Manning coefficient either "$m^{-1/3}$ " or "$s/[m^{1/3}]$".

L266 – Table 2 shows the opposite, that means CM: 15.4% & LSPIV: 8.1%. Please check.

L267 – I recommend using the same unit (do not convert) as in the table (i.e., 0.193 m instead of 19.3 cm). It makes it easy to quickly identify the value in the table.

L272 – Change to ", with the lowest values of LSPIV (6.4%) and water levels (0.063 m) for 0.015 $s/[m1/3]$ and 0.014 $s/[m^{1/3}]$, respectively".

LL273 – Don't highlight the row in the table, use a subscript (eg *) and add a sentence below the table like "* the selected optimal roughness coefficient".

L275 – I think the validation by visual analysis is not as robust as a quantitative method. However, you can justify and discuss the reasons the selected method. This is important.

L281 - I don't see a discussion of the calibration and validation processes. He mentioned in the summary that there is a need for more on-site monitoring in the future, but why is not covered in this section. It is important to do it.

L402 – The discussion must be before the conclusion section. Also, if you are adding a discussion section, remove "discussion" it from section 3 (3 Results and Discussion, L243).

Figure 8 – Missing a parenthesis.

In general, italics are used to reference figures and tables. I don't recall this being part of the journal's formatting and editing requirements. Remove the italics. Also, the table formats are not consistent (.eg., Table 1 vs Table 2). Sections and subsections do not follow the same format, some of them are case sensitive sentences, others are not (e.g., 3.1 vs 3.2) the same goes for figures and tables (L273 vs L296).

---

## Author Response (AR2)

The authors would like to express utmost gratitude for the time taken by the reviewers to assess and provide recommendations on how the manuscript can be improved. We have looked through each comment and have made changes as recommended with respect to all 31 comments. Below we list all the changes that have been made against the recommendations.

***Reference** refers to the line number in the updated track changes document.

| Reviewer Comment | Action Taken | Reference * |
|---|---|---|
| Lines 17-18. "Traditional methods ... contemplated" The sentence is vague. (e.g. the meaning and objective of "river monitoring" is not defined (and suggested to be replaced with discharge estimation at other places?).) | We agree and have taken up the suggestion to replace the term 'river monitoring' with discharge estimation | Line 18 |
| Line 18: "this UAV-system" is used without a clear definition. | We have added a brief description of the UAV-system to aid the reader to understand what we refer to. | Line 20-21 |
| Line 19: "hence probably a more accurate flow discharge" relationship between "UAV-system" and "flow discharge" would be needed to be explained. | A sentence has been added prior to (L19) describing the UAV system | Line 19 |
| Line 20: "accuracy is discharge" --> "accuracy in discharge"? | Changed from 'is' to 'in' | Line 22 |
| Lines 70-72: The sentence needs to be clarified. (not limited but "the high resolution of UAVs", UAV is a platform and high resolution maybe the photo taken from, or the DEM provided from the photo, but not a direct product of UAV") | Statement has been corrected to clearly show that the high resolution is in reference to the images, not the UAV itself | Lines 78-79 |
| Lines 109, consider removing the parenthesis if the meaning does not change. | Parenthesis have been removed | Line 120 |
| Line 258-259, I think just putting labels "a" and "b" does not justify the separation of calibration and validation. | We have removed surface velocity from the validation process seeing that the justification was not adequate. | Line 265 |
| Table 1: How was the discharge" obtained? (using ADCP? not clearly described in the method section where "flow measurement" had mentioned.) | In line 133 of the methods section we have clarified that the flow measurements were based on an ADCP | Line 143 |
| The title of section 3 is "Results and discussion". After the concluding section, a new section without a number is added. I'm not sure if such a style is standard but I'm wondering if the discussion would be better before concluding the manuscript. | We have corrected the placement error by swapping around the position of discussion and conclusion | Lines 385-410 |

| | | |
|---|---|---|
| Maybe, re-editing the section 3 (as well as editing the section 4 reflecting the discussion) would be better to do. | | |

| Reviewer Comment | Action Taken | Reference * |
|---|---|---|
| L38 – I do not think that your study has applied low-cost technology to river monitoring. In-situ monitoring is cheaper. However, I understand that compared to other systems (for example, a plane with Lidar), yours is more economic. I think it would be good to clarify this in the sentence. Same in L361) | We have clarified in both occasions that the study provides insight into the use of advanced technologies (UAV and RTK GNSS devices) for river discharge estimation. We have removed the suggestion that the method is of lower cost than in-situ estimation | Lines 44-45 |
| L40 – Here you talk about the validation process, but it is not very clear what it includes. It is worth adding a sentence explaining the methods (e.g., how calibration and validation were done). | We have added a sentence as suggested clarifying how calibration and validation were conducted. | Lines 46-47 |
| L41- While the specific objectives are important to understanding the research, I do not think it is worth mentioning them in the abstract. It would be better to mention your overall aim. This will also give space to include the methods to understand the summary of the results. | We have removed the specific objectives as suggested and have added a brief summary of the method. | Lines 29-32 |
| L107 – Please provide a summary of the method of Samboko et al. (2022) and then use the reference. If this is related to L110 please make it clear. Although it looks like L110 is related to Alvarez (2018). | A summary of the method used in Samboko (2022) has been added as suggested. | Lines 116-119 |
| L136 – Use of a single set of parentheses "(D3DFM, Deltares, 2020)". | We have now placed all the terms within one set of parentheses | Line 148 |
| L157 – I don't see the point of using a subsection (also make it as "2.4.1" if you are including it). | We have removed the subsection as advised | Line 169 |

| | | |
|---|---|---|
| L160 – Same comment on Samboko et al. 2022. Include a summary here or above (L107). | A summary was added as per suggestion on (L107)* | Lines 116-119 |
| L214 – Removed the extra dot before referring to Moriasi et al. (1983) | Error was noted and dot has been removed | Line 226 |
| L219 – This sentence should not be in bold. | The sentence was indeed not supposed to be in bold. This has been corrected | Line 233 |
| L220 – Tabulation. Maybe this is related to document conversion, please check final draft of the PDF before submitting. | Tabulation error has been noted and rectified | Line 234 |
| L217 – Tabulation errors (same as previous comment). | Tabulation error has been noted and rectified | Line 234 |
| L266 – Superscript error "-/13". Also use for the Manning coefficient either "m-1/3 " or "s/[m1/3]". | Error has been corrected | Line 281 |
| L266 – Table 2 shows the opposite, that means CM: 15.4% & LSPIV: 8.1%. Please check. | Swapping error has been corrected | Lines 281-282 |
| L267 – I recommend using the same unit (do not convert) as in the table (i.e., 0.193 m instead of 19.3 cm). It makes it easy to quickly identify the value in the table. | Reverted to same unit as suggested. Using 0.193m now | Line 282 |
| L272 – Change to ", with the lowest values of LSPIV (6.4%) and water levels (0.063 m) for 0.015 s/[m1/3] and 0.014 s/[m1/3], respectively". | Sentence has been adjusted as suggested | Line 287 |
| LL273 – Don't highlight the row in the table, use a subscript (eg *) and add a sentence below the table like "* the selected optimal roughness coefficient". | Highlight has been removed and subscript has been applied instead. | Line 279 |
| L275 – I think the validation by visual analysis is not as robust as a quantitative method. However, you can justify and discuss the reasons the selected method. This is important. | A brief discussion and justification has been added to the text as advised. | Lines 300-304 |
| L281 - I don't see a discussion of the calibration and validation processes. He mentioned in the summary that there is a need for more on-site monitoring in the | A brief discussion on the calibration and validation process has been added to the text | Lines 291-295 and Lines 301-304 |

| | | |
|---|---|---|
| future, but why is not covered in this section. It is important to do it. | | |
| L402 – The discussion must be before the conclusion section. Also, if you are adding a discussion section, remove "discussion" it from section 3 (3 Results and Discussion, L243). | We have swapped the position discussion and conclusion as suggested. The result and discussion title has also be appropriately renamed | Lines 385-410 |
| Figure 8 – Missing a parenthesis. | Parenthesis has been added | Line 630 |
| In general, italics are used to reference figures and tables. I don't recall this being part of the journal's formatting and editing requirements. Remove the italics. Also, the table formats are not consistent (.e.g. Table 1 vs Table 2). Sections and subsections do not follow the same format, some of them are case sensitive sentences, others are not (e.g., 3.1 vs 3.2) the same goes for figures and tables (L273 vs L296). | All italics on the figures and tables have been removed. All other inconsistencies have been corrected | |

---

## Author Response (AR3)

Response to reviewer comments for Manuscript titled: **Towards Affordable 3D Physics-Based River Flow Rating: Application Over Luangwa River Basin**

| Comment | Changes |
|---|---|
| With the next revision, please rename the section "Annex" to "Appendix". | Changes noted and applied as suggested |
| Please ensure that the colour schemes used in your maps and charts allow readers with colour vision deficiencies to correctly interpret your findings. Please check your figures using the Coblis – Color Blindness Simulator (https://www.color-blindness.com/coblis-color-blindness-simulator/) and revise the colour schemes accordingly. | We have edited the images to become sensitive to colour blind readers as suggested. |
| Read entire text to check grammar | Noted and checked in full. |